# Liquid water contains the building blocks of diverse ice phases

Bartomeu Monserrat [1,2], Jan Gerit Brandenburg[3,4], Edgar A. Engel[2] & Bingqing Cheng [2,5✉]

Water molecules can arrange into a liquid with complex hydrogen-bond networks and at least 17 experimentally confirmed ice phases with enormous structural diversity. It remains a puzzle how or whether this multitude of arrangements in different phases of water are related. Here we investigate the structural similarities between liquid water and a comprehensive set of 54 ice phases in simulations, by directly comparing their local environments using general atomic descriptors, and also by demonstrating that a machine-learning potential trained on liquid water alone can predict the densities, lattice energies, and vibrational properties of the ices. The finding that the local environments characterising the different ice phases are found in water sheds light on the phase behavior of water, and rationalizes the transferability of water models between different phases.

[1] Department of Materials Science and Metallurgy, University of Cambridge, 27 Charles Babbage Road, Cambridge CB3 0FS, UK. [2] Cavendish Laboratory, University of Cambridge, J. J. Thomson Avenue, Cambridge CB3 0HE, UK. [3] Interdisciplinary Center for Scientific Computing, University of Heidelberg, Im Neuenheimer Feld 205A, 69120 Heidelberg, Germany. [4] Digital Organization, Merck KGaA, Frankfurter Str. 250, 64293 Darmstadt, Germany. [5] Accelerate Programme for Scientific Discovery, Department of Computer Science and Technology, 15 JJ Thomson Ave, Cambridge CB3 0FD, UK. ✉email: bc509@cam.ac.uk

The number of experimentally observed and theoretically predicted phases of water seems to be ever growing[1]. The most ubiquitous phase on Earth, liquid water, has many intriguing properties, including a density maximum at 4 °C and ambient pressure, volume expansion upon freezing, unusually high surface tension, melting, and boiling point[2]. Liquid water exhibits no long-range order and its local structure is difficult to quantify and yet intricately related to its unique properties[2–5]. Beside the liquid, the various ice phases in the complex phase diagram of water are made from distinct local atomic environments[1], which lead to a large spread in their densities, lattice energies, and other thermodynamic as well as kinetic properties[1,6]. Apart from the direct connection with physical properties, the local structures in water are also related to the transition paths between the phases[7,8].

One intriguing question thus is the structural relationship between the different ice phases, and between ice and liquid water. This is not an easy topic to investigate, however, due to the structural complexity of the liquid[4,9,10] and the large number of ice phases[1]. In this work, we exploit state-of-the-art advances in machine learning (ML) for chemistry and materials, in order to compare the local environments in various phases of water in a general and systematic manner. More specifically, we first curate a dataset consisting of 54 representative phases of ice including all the known phases (see "Methods" section), whose densities range from 0.7 to 1.4 g/mL. Then we demonstrate that the local atomic environments found in liquid water cover the ones observed in all these ice phases, using a universal and automated framework for comparing the local similarities. As a consequence of this inclusion, a machine-learning potential (MLP)[11] that is only trained on liquid water accurately reproduces the ice properties including lattice energies, mass densities, and phonon densities of states.

## Results

**Curated dataset of diverse water phases.** We first select representative atomistic configurations of diverse crystalline and liquid phases. We start from 57 ice crystal structures, which include all the experimentally known ices. These were screened from an extensive set of 15,859 hypothetical ice structures using a generalized convex hull construction (an algorithm for identifying promising experimental candidates)[12,13]. After rigorous geometry optimizations at zero pressure (see "Methods" section), we eliminate three defective phases and the very high pressure phase X, and added the originally missing ice IV. The "Methods" section describes the dataset of the remaining 54 ice phases in more detail. Note that some structures (with particular hydrogen arrangements) represent both a proton-ordered and a proton-disordered form: for example, one ice structure prototypes both ice Ih and XI. We consider the respective minimum potential energy configurations of the ice phases, because they provide reasonable and reproducible approximations to the physical properties of ice, and serve as starting points for computing thermodynamic properties.

Compiling a set of representative structures for liquid water is less straightforward, since the liquid persists over a wide range of temperatures and pressures. We consider 1000 diverse 64-molecule snapshots of liquid water, which have previously served in training a recent MLP[11]. They were originally prepared using a three-step process. Bulk liquid systems of 64 water molecules were first equilibrated at high temperatures and densities between 0.7 and 1.2 g/mL. The resulting (de-correlated) configurations were then quenched using a steepest decent optimization. Finally, the 1000 most structurally diverse structures were extracted from all the collected liquid configurations using a farthest point sampling algorithm. There are two reasons why the 1000

configurations are representative of liquid water. First, they were constructed in order to cover a large part of the configurational space of possible atomic environments in liquid water. Second, the MLP trained using these structures reproduces many properties of water very well, including the density isobar and radial distribution functions at ambient pressure[11], which means that the training set contains the necessary information for describing liquid water at ambient pressure in a data-driven manner.

**Direct comparison of the local environments.** We employ the smooth overlap of atomic positions (SOAP)[14] local descriptors to represent the atomic environments (i.e., the displacements of all the neighbors within a cutoff radius $r_c$ around the central atom). More details regarding the representations are provided in the "Methods" section. For each structure, we then compute its global descriptors by taking the average of the local ones of all the atomic environments in that structure[15]. As the global descriptors are high-dimensional, we use principal component analysis (PCA) to build a two-dimensional embedding to visualize the relative difference (i.e., distances) between the structures. Essentially, a PCA map is the linear projection that best preserves the variances of the high-dimensional Cartesian distances of the dataset. Because only linear operations are involved throughout, the local and the global descriptors can be meaningfully projected onto the same PCA map.

We use these methodologies to analyse the 54 ice phases and the 1000 snapshots of the liquid. Figure 1a shows the PCA map of the global descriptors of all the structures: similar structures stay close on this map, while distinct ones are farther apart. The horizontal principal axis is strongly correlated with density, suggesting that density variance is a dominant feature of the

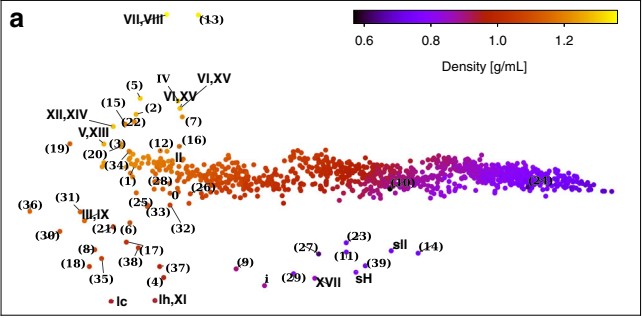

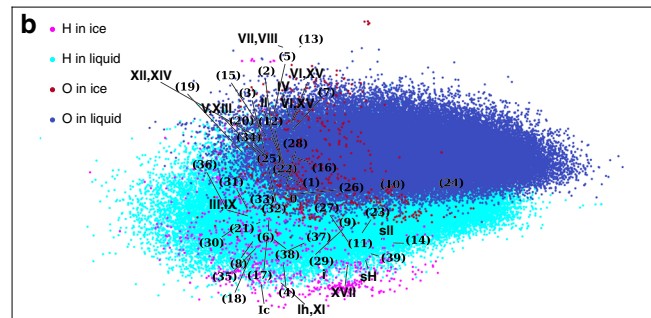

**Fig. 1 PCA maps for the 54 ice phases and the 1000 liquid water configurations.** The geometries of the ice phases have been optimized by HSE-3c. If known, each ice structure is labeled by the name of the proton ordered and disordered phase, and otherwise by a number. **a** Each dot indicates a structure, which is described by global descriptors. **b** Each small dot indicates the projection of a oxygen or hydrogen-centered local environment.

dataset. The ice phases and the liquid structures are separated on the map, while ice structures that are commonly considered to be similar (e.g., ice Ic and Ih) stay close together. The distinction between ices and liquids in the PCA map is to be expected considering the absence of long-range order in the latter. More structure–property relationships may be extracted from this PCA map, and we provide an interactive explorer of the ice and liquid water dataset in the Supplementary Data 1 ("Supplementary Data 1"), which runs in web browser and is made using Chemiscope[16].

However, the key message here is the comparison between the PCA map of the whole structures (Fig. 1a) and the one of individual atomic environments (Fig. 1b). The latter is the projection of local descriptors of all the atomic environments ($r_c = 6$ Å), onto the same PCA map as in Fig. 1a. Contrary to the clear distinction between ice and liquid shown in Fig. 1a, the local environments in liquids and ices are similar on Fig. 1b. Furthermore, the crystalline nature of the ices leads to comparatively few distinct atomic environments, and they are almost completely covered by the continuum realized in liquid water. In other words, liquid water in simulations prototypes all atomic environments pertinent to the 54 ice phases.

**Predictions of the MLP on ices.** The PCA maps provide a simple and general way of comparing and understanding the structural similarities, but choice of the representations and the linear dimensionality reduction inevitably lead to information loss and distortion. As an alternative similarity comparison, we explore how well a MLP[11] that is only trained on reference calculations for liquid water configurations describes diverse crystalline phases.

This MLP is based on revPBE0[17,18] hybrid functional density functional theory (DFT) calculations with the semiclassical D3 dispersion correction[19]. The training set contains 1593 configurations: the first 1000 are classical configurations as described above, the remaining 593 originate from path-integral molecular dynamics (PIMD) simulations at ambient conditions. We omit those PIMD configurations in the PCA analysis above because nuclear quantum effects[20] complicate the direct comparisons with

classical water, and also because those configurations had a very minor effect on the training of the MLP in our previous work[11]. The MLP uses an artificial neural network constructed according to the framework of Behler and Parrinello[21]. The total energy of the system is expressed as the sum of the individual contributions from the atom-centered environments of radius 6 Å.

Crucially, the success of the MLP hinges on the notion of "nearsightedness": energy and forces associated with a central atom are largely determined by its neighbors, and the long-range interactions can be approximated in a mean-field manner without explicitly considering the far-away atoms. This notion underlies many atomic and molecular force-field as well as most common MLPs[22]. From this point of view, to capture the energetics and dynamics of a phase of water, the key is to predict the local atom-centered contributions to the total energy and forces (Generalizations to systems with nontrivial long-range interactions like strong ionic liquids have to be done carefully as those might pose a challenge to the "nearsightedness".). In practice, this means the training set of the MLP needs to contain the essential local atomic environments of the particular phase. Following this logic, we postulate that, if the liquid water contains all the local environments of the ice phases, an MLP trained exclusively on snapshots of liquid water should also be able to describe the ice phases.

To verify our hypothesis, we benchmark the performance of the MLP against reference DFT calculations and experimental results (see more details in "Methods" section). The DFT references comprise (i) revPBE0-D3 using CP2K with similar numerical settings as the calculations performed to generate the training reference of the MLP, (ii) revPBE0-D3 using VASP and converged numerical settings, and (iii) the screened exchange hybrid HSE-3c as implemented in CRYSTAL17. Note that the multiple DFT references also provide an estimate on the intrinsic errors in these DFT calculations due to the choices on the specific hybrid functionals, numerical settings and the use of different software packages.

In Fig. 2a we show the comparison between the lattice energies and the densities (Fig. 2b) of the 54 ice phases. Note that for the

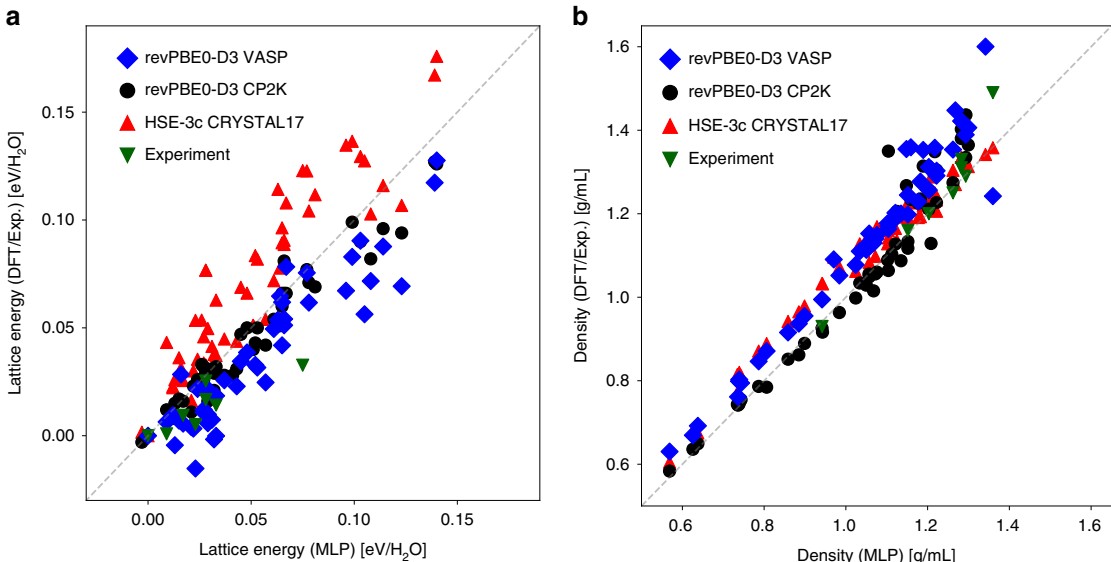

**Fig. 2 A comparison between the lattice and energy of the 54 ice phases.** The *x*-axes show the values computed using the MLP based on revPBE0-D3. **a** Results for lattice energy, and **b** shows the values for the densities of the ice phases. In both panels, the *y*-axes show the results from DFT calculations using the revPBE0-D3 functional employing VASP, the revPBE0-D3 functional employing CP2K, and the HSE-3c functional employing CRYSTAL17. The experimental references are taken from ref. [59]: the densities were measured at between 0 and 90 K and ambient pressure, and the experimental lattice energies were extrapolated to 0 K, and zero-point energies were removed. Source data are provided as a Source Data file.

lattice energies different theories or experiments have different baselines, so for each set of calculation or measurement we use the energies the ice Ih/XI structure as a reference. In general, we find an excellent agreement between the MLP predictions and all four references, particularly for the densities. In particular, the differences between the MLP results and the ab initio references are on par with the DFT differences introduced by the details of the first principles calculations. For instance, the Pearson correlation coefficient $R$ between the MLP densities and the CP2K values is 0.96 and the root mean square error (RMSE) is 0.06 g/mL, and the corresponding metrics between CP2K and VASP are 0.96 and 0.08 g/mL. For the lattice energies, the $R$ and RMSE between the MLP and the CP2K predictions are 0.95 and 9 meV/$H_2O$, respectively, compared with 0.91 and 15 meV/$H_2O$ between the CP2K and the VASP values.

The curvature of the potential energy surface around a local minimum relates to the harmonic frequencies at which the atoms in a crystal vibrate. To investigate the performance of the MLP for this quantity, we have calculated the phonon frequencies for the considered ice structures using the MLP as well as for a subset of the ice structures using revPBE0-D3 DFT calculations with both VASP and CP2K. Figure 3 provides a detailed comparison of the phonon density of states (DOS) for three representative structures with distinct densities. Figure 3a is for the ice VI phase and its proton-ordered counterpart, XV. Figure 3b corresponds to a structure that represents the ice Ih phase as well as the proton-ordered XI. Figure 3c shows the results for a low-density hypothetical phase. All the DOS show excellent agreement in both the low-energy region, corresponding to long-range dispersive crystal vibrations, and the high-energy region, corresponding to localized molecular vibrations. The small shift of low frequency phonons may be induced by the lack of long-range interactions of the MLP. A comparison across all structures can be found in the Methods, and it provides remarkable agreement between the MLP and the first principles methods across all structures in the entire energy range of vibrations.

## Discussion

The similarities between the local environments in solid and liquid phases shed light on the structure of liquid water. There have been many efforts to develop a molecular understanding of water, in terms of orientational and translational order[4], hydrogen bond networks[9] and spontaneously forming dendritic voids[10]. Our approach of using local environments observed in ice as landmark points is a new way of interpreting liquid water as a mixture of ice structures. It is worth noting that, almost as the other side of the same coin, the idea of inferring the long-range ice order from good water potentials has been discussed by Rice and coworkers in the 1980s[23,24].

On the flip side, the similarity also suggests that the liquid and ice structures are distinct in Fig. 1a not because of the difference in local environments, but due to the presence of long-range order. The conclusion that liquid water contains all the ice environments explains why the MLP trained on liquid describes the ice phases well. This generalization is not specific to this MLP. Indeed, many water models, such as the coarse grained mW[25] model, the empirical water models SPC[26] and the TIPnP series[27,28], the polarizable AMOEBA[29], and the MB-pol water potential[30,31] that are fitted to ab initio reference, qualitatively reproduce large parts of the phase diagram[23,32,33], despite having been developed primarily to simulate the liquid phase. In particular, MB-pol correctly reproduce the properties of water from the gas to the condensed phases[31]. For the MLP used here, the melting point of ice Ih and the relative stabilities between Ih and Ic have been computed[11].

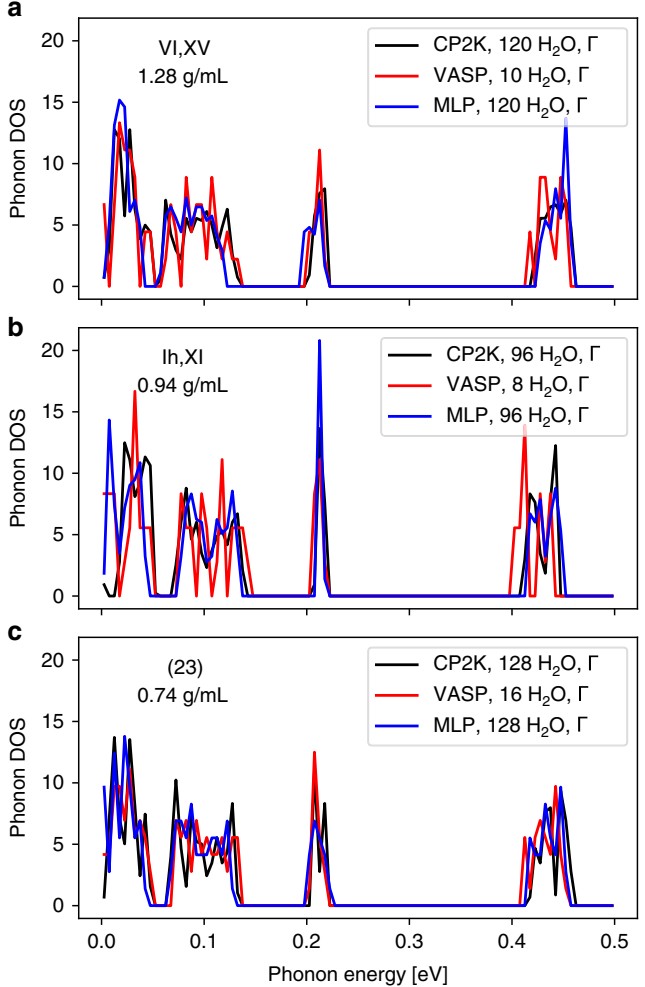

**Fig. 3 A comparison between the phonon density of states (DOS) for three ice structures. a–c** Results for three indicated ice phases with distinct densities. The three sets of DOS in each panel were computed using revPBE0-D3 DFT calculations employing VASP, revPBE0-D3 DFT calculations employing CP2K, and the MLP, all at the Gamma ($\Gamma$) point. The densities predicted at the MLP level are indicated. Source data are provided as a Source Data file.

In addition, the general and agnostic comparison of the local environments can be easily extended to study amorphous ice[34], interfaces, and water under confinement[35]. It is also interesting to investigate how nuclear quantum fluctuations[35–38] influence the distribution of the atomic environments in various phases.

Furthermore, our results illustrate the immense promise of employing MLPs in materials modeling. For instance, the MLP used here provides an accurate description of the static and vibrational properties of ice phases at a fraction of the cost of the corresponding DFT calculations. For example, the DFT calculation for the $\Gamma$-point phonons of a 4-molecule structure takes 264 CPU hours, and that for a 52-molecule structure takes 16,000 CPU hours, compared with just a few minutes for both on a laptop using the MLP. Besides, the fact that one can only train a MLP on one liquid phase and apply the potential to other phases evince the extend of its "extrapolability", which significantly facilitates the constructions of the potentials. It is worth mentioning that, the MLP for water is stable enough to run MD and PIMD for all the phases. Despite the success of the MLP trained on the liquid water, we want to caution that for systems where long-range interactions are important, such as strongly ionic

systems, the short-range nature of the MLP may pose a limit on the overall accuracy. For a given system, it is also not certain if its melt contains sufficient local ordering to reproduce the atomic environments found in the solid phases.

Last but not least, using MLPs as tools for comparing atomic environments offers a new approach of analyzing complex atomic systems in an agnostic and general manner. Our analysis is, of course, not restricted to the chosen DFT level and can be extended to incorporate new developments in the field of ab initio methods[39–42].

To summarize, we compare the local environments in various crystalline ice phases and liquid water, using two ML-based approaches. We demonstrate that liquid water contains all the local atomic environments in diverse ice phases. Our conclusion provides a new and fundamental perspective on the understanding of liquid water and ices, and guides future efforts for modeling water.

## Methods

**SOAP representations for atomic environments**. Numerous representations of atomic environments have been developed[14,43–45], and here we use the SOAP representation[14]. SOAP encodes the local environment $\mathcal{X}$ around a central atom using a smooth atomic density function

$$\rho_{\mathcal{X}}^{\alpha}(\mathbf{r}) = \sum_{|r_i| < r_c} \exp\left(-\frac{[\mathbf{r} - \mathbf{r_i}]^2}{2\sigma^2}\right), \quad (1)$$

by summing over Gaussians centered on each atom $i$ of species $\alpha$ (here hydrogen or oxygen) within a given cutoff distance $r_c$ of the central atom. The density $\rho_{\mathcal{X}}^{\alpha}(\mathbf{r})$ is then expanded in a basis of orthonormal radial functions $g(|r|)$ and spherical harmonics $Y_{lm}(\hat{r})$ as

$$\rho_{\mathcal{X}}^{\alpha}(\mathbf{r}) = \sum_{nlm} c_{nlm}^{\alpha} g_n(|r|) Y_{lm}(\hat{r}). \quad (2)$$

Finally, the power spectrum is taken as

$$k_{nn'l}^{\alpha}(\mathcal{X}) = \sqrt{\frac{8}{2l+1}} \sum_m (c_{nlm}^{\alpha})^* c_{n'lm}^{\alpha}, \quad (3)$$

which characterizes $\mathcal{X}$ in a translation, permutation, and rotation invariant form[14,15]. The vector $\{k_{nn'l}^{\alpha}\}$ constructed in this way up to certain cutoffs $l_{max}$ and $n_{max}$ can then be used as the local fingerprint $\Psi(\mathcal{X})$. We set the radius of the atomic environment to be $r_c = 6$ Å, so that it includes the second hydration shell of water molecules, and expand the SOAP descriptor up to $l_{max} = 6$ and $n_{max} = 6$. PCA maps that were constructed using SOAP with different hyper-parameters are in the Supplementary Figs. 1 and 2. In practice, we use the DScribe Python package for constructing descriptors[46], and the ASAP Python package for the subsequent analysis (provided as Supplementary Software 1).

**Choice of the ice configurations**. The initial 57 structures are based on an extensive survey of ice[13] that generated 15,859 configurations, by exploiting the isomorphism between ice, experimentally known zeolites and theoretically-enumerated four-connected SiO$_2$ networks. The resulting ice-like configurations were subsequently locally relaxed, before a generalized convex hull construction[12] was employed to screen for the ice structures that may be stable under certain thermodynamic conditions. These structures include the experimentally known phases of ice except for ice IV, which we then add back into the selection. Figure 4 shows the PCA map of the locations of the selected phases. Notably, many ice

phases come in pairs of a low-temperature proton-ordered form and a higher-temperature proton-disordered form: Ih and XI, III and IX, V and XIII, VI and XV, VII and VIII, and XII and XIV. In this work, we focus on the particular proton-ordered realizations of these phases made available with ref. [13].

**Initial geometry optimization of the ice structures using HSE-3c**. In ref. [13] the ice structures were optimized at the PBE DFT level of theory using a coarse $k$-point grid and plane wave basis, trading accuracy for computational efficiency. For this study, we have therefore performed well-converged geometry optimization for the structures, by running a few cycles of local geometry optimizations followed by identifying and imposing crystal space group symmetries. These local optimizations are performed with the screened exchange hybrid functional HSE-3c[47] using tight optimization thresholds as implemented in CRYSTAL17[48,49]. The Brillouin zone is sampled with a Γ-centered Monkhorst-Pack grid that has been converged individually for every system to yield a lattice energy accuracy well below 1 meV. HSE-3c has been shown to yield excellent molecular and intermolecular geometries as well as good noncovalent interaction energies[50–52], and in particular, suitable for water and ices. As all ice structures considered here were variable-cell geometry optimized at zero pressure, we were able to directly compare them in the subsequent analysis. Note also that the proton arrangements are typically "locked in" during optimizations, so the proton (dis) ordering of each optimized ice structure is determined by the initial configuration and does not necessarily represent the ground state. These relaxed geometries of the 54 ice phases, together with their distributions of the oxygen–oxygen distances and the oxygen–oxygen–oxygen angles are included in the Supplementary Figs. 3 and 4.

**Geometry optimization using VASP**. The geometries of all ice structures were further refined with revPBE0-D3 using the VASP package[53,54]. The equilibrium volumes critically depend on the energy cutoff, and we used a relatively high value of 1200 eV to obtain converged results. The $\mathbf{k}$-point sampling grids were the same as those determined for the HSE-3c calculations described above. Structures were constrained to their initial symmetries throughout the geometry optimization, and convergence was achieved with forces below $10^{-3}$ eV/Å and stress components below $10^{-2}$ GPa.

**Geometry optimization using CP2K**. We computed the equilibrium densities and the lattice energies of the ice structures using the CP2K code[55] with the revPBE0-D3. The computational details of the calculations are identical with refs. [11,56], although the planewave cutoff energy was increased to 800 eV, to obtain smooth volume-energy curves. Despite a considerable amount of effort, the geometry optimization for three structures did not converge to reasonable values, so these calculations were discarded.

**Phonon calculations**. For the phonon calculations using the MLP, we first computed the Hessian matrix for the 54 geometry-optimized ice phases using finite displacements of 0.01 Å of each atom from its equilibrium position along $x$, $y$, and $z$ axes. Then the Hessian matrix was diagonalized to obtain the phonon frequencies as the square root of the eigenvalues. We performed those phonon calculations for the ice systems in both their original cell taken from ref. [13], and in supercells of this original cell, obtained by repeating the original cell along all three crystallographic directions so that each dimension of the supercell is longer than 8 Å.

The phonon calculations using CP2K follow the same approach, and the DFT settings are identical to those for geometry optimization. Presumably due to numerical issues of the specific DFT setup that we used (e.g., CP2K only supports Γ-point sampling for hybrid functionals), a number of the ice phases contain imaginary phonons at the Γ-point, even after several rounds of geometry optimization. We discarded the CP2K phonon DOS for these phases, and only show the ones with real frequencies in Fig. 5.

The VASP phonon calculations were performed using the structures optimized with VASP with the same parameters described above. We used the finite displacement method[57] in conjunction with nondiagonal supercells[58], and commensurate $\mathbf{k}$-point grids were used to sample the electronic Brillouin zones of the supercells. The Hessian of a given nondiagonal supercell was calculated by displacing each atom from its equilibrium position by 0.01Å in symmetry-inequivalent directions and calculating the force constants by finite differences. The dynamical matrix for a given $\mathbf{q}$-point grid of the vibrational Brillouin was determined by combining the results from multiple nondiagonal supercell calculations as described in the ref. [58]. The resulting dynamical matrix was diagonalized to obtain the phonon frequencies and eigenvectors.

In all sets of phonon calculations, imaginary phonons appear in multiple structures at various $\mathbf{q}$-points in the Brillouin zone. This reflects the fact that the protons in many ice structures are disordered, and when we attempt to model them as periodic ordered structures using the unit cells from ref. [13], we are artificially constraining them to a saddle point of the potential energy surface rather than to a local minimum. In some of the structures, instabilities appear even at the Γ-point, and this is caused by the symmetrization step in preparing the structures, which again can place them at a saddle point. The imaginary phonons in this case break some of the imposed symmetries to lower the energy. These problems can be resolved by replicating the original simulation cell and re-relaxing the atomic positions of the supercell to allow for the appearance of disorder that lowers the overall energy, or by re-relaxing the primitive cell without imposing symmetry in

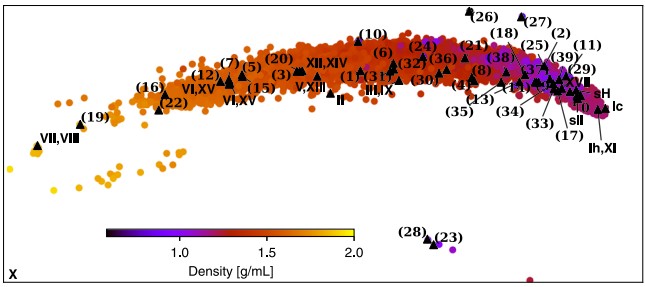

**Fig. 4 PCA Map for 15,859 ice phases.** The phases selected for the current study, as well as the phase X (outside the map), are marked on the map.

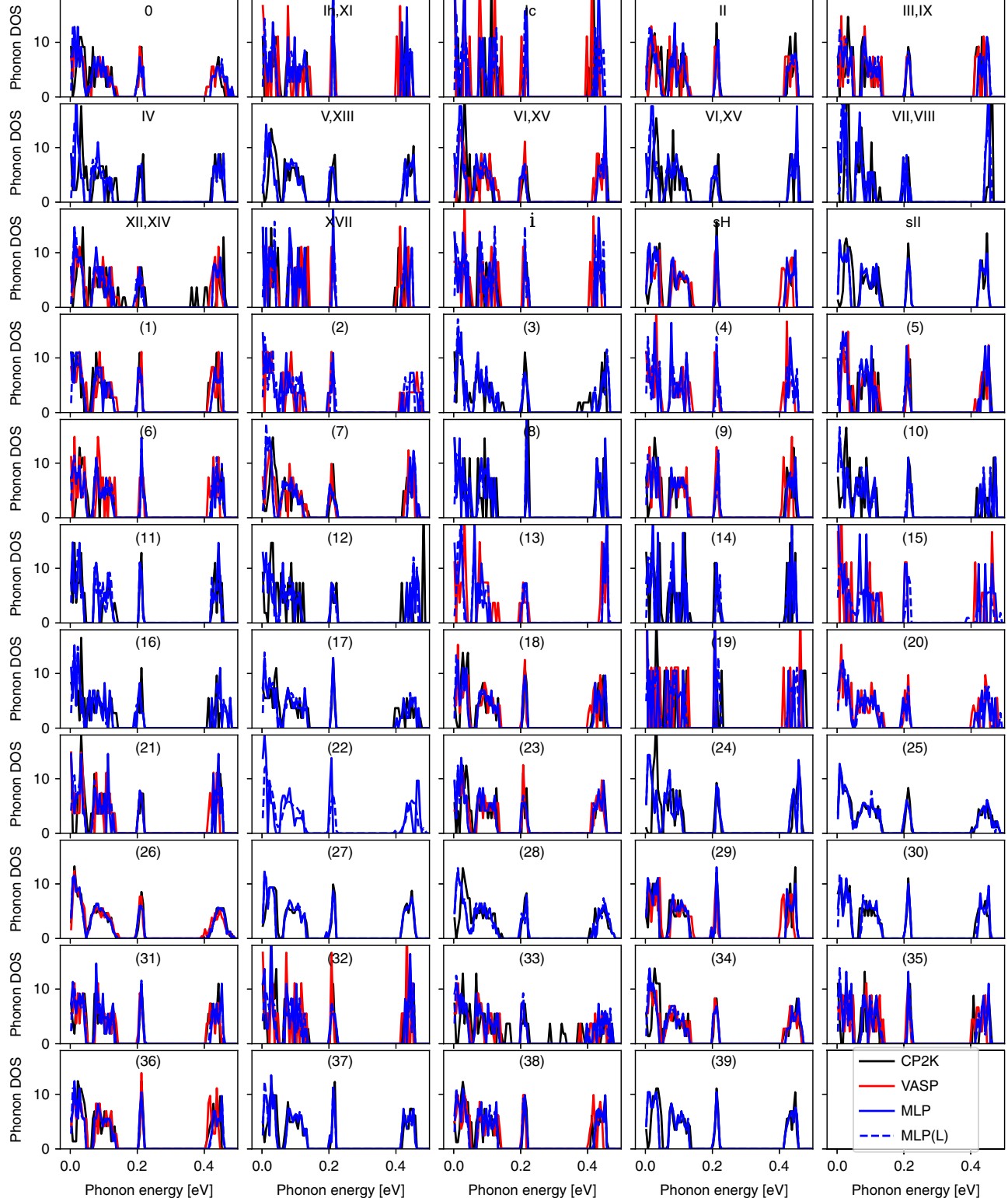

**Fig. 5 A comparison between the phonon DOS at the Gamma point for 54 ice phases.** The DOS were computed using revPBE0-D3 DFT calculations employing VASP, revPBE0-D3 DFT calculations employing CP2K, MLP using the original supercell, and MLP using replicated cell (MLP(L)). For the VASP and CP2K calculations, only those that exhibited no imaginary phonons are shown here. Source data are provided as a Source Data file.

the case of $\Gamma$-point phonons. This additional step is computationally trivial for the MLP calculations, but computationally extremely costly for the hybrid functional calculations using DFT. As a consequence, it is computationally prohibitive to accurately calculate the phonon densities of states for all ice structures at the DFT level, and we only consider a subset in Fig. 5.

**Data availability**

The 54 ice structures correspond to classical 0 K structures without external pressure, the CRYSTAL17, CP2K, and VASP input files, the Python notebook for analysis are provided in the Supplementary Data 1 ("Supplementary Data 1"), and available on https://github.com/BingqingCheng/ice-in-water. Source data are provided with this paper.

## Code availability

We used the ASAP code for most of the analysis, which is available at: https://github.com/BingqingCheng/ASAP The Python notebook for making Fig. 1 is included in the Supplementary Data 1 ("Supplementary Data 1").

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

## Acknowledgements

We thank Alexandre Tkatchenko, and Francesco Paesani for critically reading the manuscript and giving helpful feedback. B.M. acknowledges support from the Gianna Angelopoulos Programme for Science, Technology, and Innovation, and from the Winton Programme for the Physics of Sustainability. B.C. acknowledges funding from the Swiss National Science Foundation (Project P2ELP2-184408). B.M. and B.C. acknowledge the resources provided by the Cambridge Tier-2 system operated by the University of Cambridge Research Computing Service (http://www.hpc.cam.ac.uk) funded by EPSRC Tier-2 capital grant EP/P020259/1. B.C. acknowledges allocation of CPU hours by CSCS under Project ID s957.

## Author contributions

B.M., J.G.B., and B.C. designed research; B.M., J.G.B., and B.C. performed research; B.C. analyzed data; B.M., J.G.B., E.A.E., and B.C. wrote the paper.

## Competing interests

The authors declare no competing interests.
