## [Peer Review File · Nature Communications]

Reviewer #1 (Remarks to the Author):

This manuscript aims to investigate the structural relationship between liquid water and different ice phases. This is an important topic of great interests to the community due to its relevance to the fundamental understanding of phase transformations in materials (e.g., ice formation/melting) and their potential industrial and technological applications. In my opinion, a data-driven approach that provides physical understanding of this structural relationship at the molecular level and in relation to experimental data for a complex and anomalous material like water would warrant publication in Nat. Commun., but the manuscript in its current form does not deliver enough insights/novelties for such recommendation.

The work presented in this manuscript is built upon the authors' previous studies (10.1038/s41467-018-04618-6 and 10.1073/pnas.1815117116). Briefly, the same methodologies are applied to obtain comparison structures (i.e., 57 ice and 1000 liquid water configurations) which the authors then analyzed using a machine learning (ML) approach. I do not have trouble with this, as the methods seem established, and the authors also refined the data in this work. Results from their first study and this study both suggest that combining SOAP descriptors and PCA maps is a robust way of identifying stable ice structures similar to those observed in experiments and separate them by structural similarity based on the local environment of atoms (neighbors within 5 - 6 Å). However, other than a notable correlation between density and one of the PCA axes, I have trouble visualizing and extracting physical insights from these plots that can advance my understanding of the structural relationship between ice and liquid water. Perhaps the authors can identify specific data points and perform additional analysis to extract more information (e.g., radial and angular distribution of the neighbors, degree of photon ordering, etc.) and provide the physical insights.

Furthermore, despite the inclusion of limited lattice energy and density data of ice phases from experiments, I feel that there could be a major disconnect between this study and water observed in experiments. As far as I understand, the machine-learning potential (MLP) is trained on liquid water configurations obtained from DFT while the ice structures are also optimized using DFT, so the match in PCA maps and the good predictions of ice properties from a MLP that is trained on just DFT liquid water data do not indicate if real liquid water contains ice-like building blocks, which the manuscript title seems to suggest. At most, this study validates that the liquid water training dataset of MLP contains ice-like building blocks, which explains why MLP has good predictions of ice properties despite training on just DFT liquid water data. Upon looking at Figure 1 of the original MLP paper published in PNAS (the authors' second study), it appears that MLP predicts liquid water densities below 240K to be quite close to experimental ice Ih densities, so I am less convinced that data predicted by this model can be directly compared with ice data from experiments.

Questions:

1. How do the authors select the neighbor cutoff in the SOAP descriptor? How sensitive are the results to the value of this cutoff? Is the same cutoff used for ice structures at different pressure?
2. There is a mention of taking the minimum potential energy configurations of proton-ordered and photon-disordered form of ice. Do the hydrogen atoms in the lowest energy photon-disordered form of ice remain disordered? Or do they become like the proton-ordered form?
3. The reference experimental data of ice phases are measured at various temperatures. How are

the temperature and pressure dependencies handled in the analysis? From the PCA results, can one infer the structural stabilities of ice phases at different temperatures and pressures?

4. Minor: Section numbers are missing in "see Sec. "

Reviewer #2 (Remarks to the Author):

The manuscript by Monserrat et al. provides an innovative analysis of the link between the local structure of liquid water and that of 53 real and hypothetical ice phases.

The authors prove that a machine learning potential based on neural networks and fitted on liquid water reproduced accurately the density, lattice energy and vibrational density of states of the ice phases. The principal component analysis of the high-dimensional global and local descriptors of the local environment of the water molecules shows that the ensemble of the local structures of ices is (roughly) contained in that of the liquid. This feature justifies why the ML potential built for water gives good results for ice as well.

The paper is very well written and the results are convincing. The methodology is solid and sufficient details and data are provided to reproduce the calculations. The results are important for the understanding of the physics of ice and water, and, beyond that, for the use of machine learning potentials in molecular simulations. For these I would recommend that the paper is published in Nature Communications, provided that the authors address the following minor issues:

1- An important and general consequence of this result is that it justifies generating interatomic potentials from liquid phases, which often entail short-range order, to represent crystalline phases with long-range order. The authors underline this point in the discussion, however, it would be better to include also a caveat for those situation in which the nearsightedness of machine learning potentials may be a limit, and to materials in which the melt is not sufficiently structured to reproduce the local environment of solids. Would this approach be expected to work for strongly ionic systems?

2- The authors mention the ability of other potentials to reproduce the phase diagram of ice and water. How about the MLP used in this work? Has it been benchmarked against the phase diagram? Should one refer explicitly to the previous work by Cheng et al. (PNAS2019) where the ML potential was generated?

Davide Donadio

We thank the referees for carefully reading our manuscript and providing constructive criticism. In response to the referees' comments, we have added new analyses and made a number of changes in the revised version of the main text (highlighted in blue) and the SI (added 4 supplementary figures and an interactive html file). In what follows we respond in detail to the individual points raised by the referees.

Reviewer #1 (Remarks to the Author):

This manuscript aims to investigate the structural relationship between liquid water and different ice phases. This is an important topic of great interests to the community due to its relevance to the fundamental understanding of phase transformations in materials (e.g., ice formation/melting) and their potential industrial and technological applications. In my opinion, a data-driven approach that provides physical understanding of this structural relationship at the molecular level and in relation to experimental data for a complex and anomalous material like water would warrant publication in Nat. Commun., but the manuscript in its current form does not deliver enough insights/novelties for such recommendation.

Authors:

We thank the referee for taking the time to review our work, and for the useful suggestions. We have clarified certain parts of the manuscript, added new analyses, and expanded our discussions.

The work presented in this manuscript is built upon the authors' previous studies (10.1038/s41467-018-04618-6 and 10.1073/pnas.1815117116). Briefly, the same methodologies are applied to obtain comparison structures (i.e., 57 ice and 1000 liquid water configurations) which the authors then analyzed using a machine learning (ML) approach. I do not have trouble with this, as the methods seem established, and the authors also refined the data in this work. Results from their first study and this study both suggest that combining SOAP descriptors and PCA maps is a robust way of identifying stable ice structures similar to those observed in experiments and separate them by structural similarity based on the local environment of atoms (neighbors within 5 - 6 Å). However, other than a notable correlation between density and one of the PCA axes, I have trouble visualizing and extracting physical insights from these plots that can advance my understanding of the structural relationship between ice and liquid water. Perhaps the authors can identify specific data points and perform additional analysis to extract more information (e.g., radial and angular distribution of the neighbors, degree of photon ordering, etc.) and provide the physical insights.

AUTHORS:

The referee is correct in pointing out that the present work is built upon our previous studies (Nat. Commun. 2018; PNAS 2019). However, although we utilized the data and the methods made available in these studies, the conceptual framework and the key conclusion of this paper are novel. Rather than identifying and analyzing stable ice polymorphs, we focus on demonstrating that liquid water contains *all* atomic environments found in ice phases. We

have rephrased the discussions about the PCA maps on page 2 to highlight these points. We also explore an important consequence of this observation, by showing that a machine-learning potential trained only using the information from the liquid, can describe the diverse ice phases.

In addition, in the Supplementary Material, we now show the distributions of the oxygen-oxygen distances, and the oxygen-oxygen-oxygen angles (reproduced below) for all the ice phases and the liquid structures. This new analysis serves two purposes: 1. It confirms the distributions in the liquid span the ones for ices, although such analysis is less general and agnostic than the PCA maps. 2. The distinct distributions from all the ice phases highlight the diversity of the considered structures.

We completely agree with the referee that our work may enable a deeper exploration of the structural relationships *between* certain ice phases using data-driven visualization techniques, although this is not the focus of the current paper, as we mainly use the PCA maps to show that liquid environments cover the ice ones. To facilitate such future efforts, however, we now

provide an interactive structure-property explorer as a part of the Supplementary Material, which is an html file that can run in any standard web browser.

Furthermore, despite the inclusion of limited lattice energy and density data of ice phases from experiments, I feel that there could be a major disconnect between this study and water observed in experiments. As far as I understand, the machine-learning potential (MLP) is trained on liquid water configurations obtained from DFT while the ice structures are also optimized using DFT, so the match in PCA maps and the good predictions of ice properties from a MLP that is trained on just DFT liquid water data do not indicate if real liquid water contains ice-like building blocks, which the manuscript title seems to suggest. At most, this study validates that the liquid water training dataset of MLP contains ice-like building blocks, which explains why MLP has good predictions of ice properties despite training on just DFT liquid water data. Upon looking at Figure 1 of the original MLP paper published in PNAS (the authors' second study), it appears that MLP predicts liquid water densities below 240K to be quite close to experimental ice Ih densities, so I am less convinced that data predicted by this model can be directly compared with ice data from experiments.

AUTHORS:

The referee expressed concerns of whether the DFT water reasonably represents water observed in experiments. The specific hybrid DFT functional (revPBE0-D3) that we based the MLP on has been carefully benchmarked on both high level theoretical data (Goerigk et al. PCCP 2017, Brandenburg et al. JCP 2019) as well as on experimental ice stabilities (Brandenburg et al. JCP 2015). Furthermore, ab initio molecular dynamics and path-integral molecular dynamics show that revPBE0-D3 water matches the radial distribution functions, diffusivity and spectroscopy of experimental water (Marsalek, Markland JPCL 2017). The close agreement with experiments with regards to the melting points of both light and heavy water, density isobar, and the temperature of maximum density are further illustrated in our previous work that introduced the MLP (Cheng et al. PNAS 2019).

The referee observed that in the PNAS 2019, liquid water densities below 240K are quite close to experimental ice Ih densities (see the figure below). This requires some clarification. We suppose the referee is referring to the crossover between the density of classical water and that of experimental ice at about 230 K.

In figure 1 of the PNAS 2019 paper (reproduced below), we show both the densities for classical water (dashed curves) and (quantum-mechanical) light water H₂O (solid curves) at ambient pressure. The experimental values also shown on the same figure are from light water. For both the classical and the light liquid, there is a density maximum at 280 K, consistent with experimental data. Overall, the MLP light liquid water is about 4% less dense than experimental liquid, and the MLP ice is about 2% less dense.

[REDACTED]

From the simulations using the MLP, the light water is about 1% denser than its classical counterpart. Meanwhile, experimental heavy, deuterated ice as well as liquid water (the “more classical” versions of their protonated counterparts) are less dense -- this is exactly what is captured by the MLP.

[REDACTED]

(Roettger et al., Acta Cryst. B50, 644-648 (1994))

As for the density comparison between ice and water at undercooled conditions, experiments found that water becomes less dense at lower temperature, and eventually supercooled protonated water approaches 0.924 g/cm^3 at around 205 K, which is the same as the density of ice at that temperature of around 0.923 g/cm^3 to within the experimental errors. This decreasing difference in densities between ice and liquid at undercooled conditions is exactly what has been captured by our MLP trained on DFT.

[REDACTED]

(Mallamace et al., PNAS 104 (47) 18387-18391 (2007), doi: 10.1073/pnas.0706504104)

To sum up, the issue raised by the referee is not against experimental observations. While there is always a gap between theoretical modelling and the physical system, the many key physical properties reproduced by the revPBE0-D3 and the MLP water suggest that they are appropriate for probing the atomic environments in ice and water. Nevertheless, we have added the phrase “in simulations” in the abstract and the results to clarify the distinction between experimental and simulated water.

Questions:

1. How do the authors select the neighbor cutoff in the SOAP descriptor? How sensitive are the results to the value of this cutoff? Is the same cutoff used for ice structures at different pressure?

AUTHORS:

In order to compare structural similarities or atomic environments, the same descriptor must be used for all structures. Thus, we describe all structures (and environments) using a SOAP description with the same hyperparameters and, specifically, a cutoff radius of 6 Angstrom. The rationale for choosing a cutoff radius of 6 Angstrom is that (i) it matches the cutoff radius employed in the MLP, and (ii) the O-O and O-H radial distribution functions for water and ice show that it includes the first and second hydration shells. Of course, the latter argument also motivated the choice of cutoff in the construction of the MLP.

Furthermore, in the Supplementary Materials we have added two sets of PCA plots that used either 4 or 8 Angstrom cutoff. Basically, the qualitative picture does not change. With the smaller cutoff (about the radius of the first hydration shell), the atomic environments in liquid water almost completely cover the solid ones, while with the larger cutoff (beyond the third hydration shell) ice environments start to look more distinct from the liquid's. This is to be expected, as the larger the cutoff, the more long-range orderings in the ice are taken into account.

We would also like to clarify that all ice structures involved in the comparison to snapshots from liquid water are zero pressure configurations. Even though some of them are derived from experimental structure data at different pressures (while others originate straight from 4-connected zeolitic networks from the databases of Treacy and Deem), all ice structures considered here are variable-cell geometry optimised at zero pressure, rendering them comparable to each other. We have added a sentence in the Methods section to clarify this.

2. There is a mention of taking the minimum potential energy configurations of proton-ordered and proton-disordered form of ice. Do the hydrogen atoms in the lowest energy proton-disordered form of ice remain disordered? Or do they become like the proton-ordered form?

AUTHORS:

The hydrogen atoms in the lowest energy proton-disordered form of ice remain disordered. This is mainly because we start from initial ice configurations that satisfy the Bernal-Fowler ice rules and not necessarily the ground state of proton configuration, and such proton arrangements are “locked in” during the subsequent geometry optimizations into local minima. We have added a sentence in the Methods section to clarify this.

3. The reference experimental data of ice phases are measured at various temperatures. How are the temperature and pressure dependencies handled in the analysis? From the PCA results, can one infer the structural stabilities of ice phases at different temperatures and pressures?

AUTHORS:

The reference experimental data has been extrapolated to 0 K as outlined in Ref. [16], i.e. the experimental densities were measured at low temperatures (≤ 90 K) and ambient pressure. The experimental lattice energies were extrapolated to 0 K, and zero-point vibrational energies were removed. As such, we can directly compare the *in silico* geometries with the measured ones. We have now added these explanations to the caption of Fig. 2.

Regarding whether one can infer the structural stabilities of ice phases from PCA maps, we did not find a robust and straightforward way of doing so for specific pressure and temperature conditions. We think this instead requires rigorous free energy estimations, which is a part of our ongoing work for a separate project.

4. Minor: Section numbers are missing in "see Sec. "

AUTHORS:

We thank the reviewer for pointing out this typographical error and have fixed it.

Reviewer #2 (Remarks to the Author):

The manuscript by Monserrat et al. provides an innovative analysis of the link between the local structure of liquid water and that of 53 real and hypothetical ice phases.

The authors prove that a machine learning potential based on neural networks and fitted on liquid water reproduced accurately the density, lattice energy and vibrational density of states of the ice phases. The principal component analysis of the high-dimensional global and local descriptors of the local environment of the water molecules shows that the ensemble of the local structures of ices is (roughly) contained in that of the liquid. This feature justifies why the ML potential built for water gives good results for ice as well.

The paper is very well written and the results are convincing. The methodology is solid and sufficient details and data are provided to reproduce the calculations. The results are important for the understanding of the physics of ice and water, and, beyond that, for the use of machine learning potentials in molecular simulations. For these I would recommend that the paper is published in Nature Communications, provided that the authors address the following minor issues:

AUTHORS:

We thank the referee for taking the time to review our work, for their insightful suggestions and for the recommendation to publish it in Nature Communications.

1- An important and general consequence of this result is that it justifies generating interatomic potentials from liquid phases, which often entail short-range order, to represent crystalline phases with long-range order. The authors underline this point in the discussion, however, it would be better to include also a caveat for those situation in which the nearsightedness of machine learning potentials may be a limit, and to materials in which the melt is not sufficiently structured to reproduce the local environment of solids. Would this approach be expected to work for strongly ionic systems?

AUTHORS:

We agree completely with the referee that long-range interactions that cannot be mapped onto effective short range descriptors of MLPs can be problematic. In particular, strongly ionic systems have non-trivial electrostatic interactions that might be prone to this issue. It is also true that the approach of learning from the liquid and generalizing to the solids is not expected to work for all materials, and have to be carefully tested on a case-by-case manner.

To highlight these points, we have now added the following to the discussion section:

“Despite the success of the MLP trained on the liquid water, we want to caution that for systems where long-range interactions are important, such as strongly ionic systems, the short-range nature of the MLP may pose a limit on the overall accuracy. For a given system, it is also not certain if its melt contains sufficient local ordering to reproduce the atomic environments found in the solid phases.”

2- The authors mention the ability of other potentials to reproduce the phase diagram of ice and water. How about the MLP used in this work? Has it been benchmarked against the phase

diagram? Should one refer explicitly to the previous work by Cheng et al. (PNAS2019) where the ML potential was generated?

AUTHORS:

Indeed we are computing the full pressure-temperature phase diagram using the MLP. This is in fact quite a lot of work, and for this reason only a few water potentials (tip4p, tip4p/ice, SPC, mW) have been used to compute the full phase diagram. In our case, the computation is even more tedious, as we further take into account nuclear quantum effects and proton orderings, and we also start from 54 phases of candidate ice phases instead of the experimental ones. We hope to submit this work soon as a follow-up of the current study. In the meantime, we have added a sentence to refer to Ref.[11]: “For the MLP used here, the melting point of ice Ih and the relative stabilities between Ih and Ic have been computed in Ref.~\cite{Cheng2019}.”

Davide Donadio

Reviewer #1 (Remarks to the Author):

The authors have carefully addressed the comments of the reviewers. In my opinion, the revised manuscript has much improved in terms of highlighting key messages of the work, providing potential caveats of the results, and clarifying technical details that were missing in the original manuscript.

I very much appreciate the authors for performing additional calculations and go a step further in providing an interactive data explorer for the readers. Although I was hoping to see more results showing strong physical insights of the identified building blocks linking ice/liquid water structures beyond the correlations inferred from 2D projected PCA maps, I understand that the topic is complex and there are many challenges to be addressed that are beyond the scope of this manuscript. The authors have laid significant groundwork for future studies and the manuscript in its current form has enough novelties that appeal to a wide range of readers.

Reviewer #1 (Remarks to the Author):

The authors have carefully addressed the comments of the reviewers. In my opinion, the revised manuscript has much improved in terms of highlighting key messages of the work, providing potential caveats of the results, and clarifying technical details that were missing in the original manuscript.

I very much appreciate the authors for performing additional calculations and go a step further in providing an interactive data explorer for the readers. Although I was hoping to see more results showing strong physical insights of the identified building blocks linking ice/liquid water structures beyond the correlations inferred from 2D projected PCA maps, I understand that the topic is complex and there are many challenges to be addressed that are beyond the scope of this manuscript. The authors have laid significant groundwork for future studies and the manuscript in its current form has enough novelties that appeal to a wide range of readers.

AUTHORS:

We thank the referee for their insightful suggestions that helped improving our work, and for the positive assessment.